# Histological Analysis of a Mouse Model of the 22q11.2 Microdeletion Syndrome

**DOI:** 10.3390/biom13050763

**Published:** 2023-04-27

**Authors:** Hidenori Tabata, Daisuke Mori, Tohru Matsuki, Kaichi Yoshizaki, Masato Asai, Atsuo Nakayama, Norio Ozaki, Koh-ichi Nagata

**Affiliations:** 1Department of Molecular Neurobiology, Institute for Developmental Research, Aichi Developmental Disability Center, 713-8 Kamiya, Kasugai 480-0392, Japan; 2Department of Psychiatry, Nagoya University Graduate School of Medicine, Nagoya 466-8550, Japan; d-mori@med.nagoya-u.ac.jp; 3Brain and Mind Research Center, Nagoya University, Nagoya 466-8550, Japan; 4Department of Cellular Pathology, Institute for Developmental Research, Aichi Developmental Disability Center, 713-8 Kamiya, Kasugai 480-0392, Japan; matsukit@inst-hsc.jp (T.M.); atsuon@inst-hsc.jp (A.N.); 5Department of Disease Model, Institute for Developmental Research, Aichi Developmental Disability Center, 713-8 Kamiya, Kasugai 480-0392, Japan; yosizaki@inst-hsc.jp (K.Y.); masato-a@inst-hsc.jp (M.A.); 6Department of Neurochemistry, Nagoya University Graduate School of Medicine, Nagoya 466-8550, Japan; 7Pathophysiology of Mental Disorders, Nagoya University Graduate School of Medicine, Nagoya 466-8550, Japan; ozaki-n@med.nagoya-u.ac.jp; 8Institute for Glyco-core Research (iGCORE), Nagoya University, Chikusa-ku, Nagoya 464-0814, Japan

**Keywords:** 22q11.2 deletion syndrome, dopamine system, schizophrenia, dopaminergic neurons, psychiatric disorders, dendrite formation, dendritic spine

## Abstract

22q11.2 deletion syndrome (22q11.2DS) is associated with a high risk of developing various psychiatric and developmental disorders, including schizophrenia and early-onset Parkinson’s disease. Recently, a mouse model of this disease, Del(3.0Mb)/+, mimicking the 3.0 Mb deletion which is most frequently found in patients with 22q11.2DS, was generated. The behavior of this mouse model was extensively studied and several abnormalities related to the symptoms of 22q11.2DS were found. However, the histological features of their brains have been little addressed. Here we describe the cytoarchitectures of the brains of Del(3.0Mb)/+ mice. First, we investigated the overall histology of the embryonic and adult cerebral cortices, but they were indistinguishable from the wild type. However, the morphologies of individual neurons were slightly but significantly changed from the wild type counterparts in a region-specific manner. The dendritic branches and/or dendritic spine densities of neurons in the medial prefrontal cortex, nucleus accumbens, and primary somatosensory cortex were reduced. We also observed reduced axon innervation of dopaminergic neurons into the prefrontal cortex. Given these affected neurons function together as the dopamine system to control animal behaviors, the impairment we observed may explain a part of the abnormal behaviors of Del(3.0Mb)/+ mice and the psychiatric symptoms of 22q11.2DS.

## 1. Introduction

22q11.2 deletion syndrome (22q11.2DS) is a genetic disorder caused by a microdeletion of that region of chromosome 22, which is estimated to occur in 1 to 4000 live births [1,2]. Most (90%) 22q11.2DS patients have a 3.0 Mb deletion encompassing 45 genes, whereas 8% have a 1.5 Mb deletion [3,4]. The individuals show cardiac malformations, velopharyngeal insufficiency, congenital hypocalcemia, thymus hypoplasia, and characteristic craniofacial structures with various extents of presentation [5]. Importantly, the 22q11.2 deletion is one of the strongest genetic risks of developing psychiatric diseases including schizophrenia, autism spectrum disorder, and early onset Parkinson’s disease [6,7]. Aiming to elucidate the etiology of 22q11.2DS, several lines of mouse models have been generated. Four previously generated mouse models mimic the less frequent 1.5 Mb deletion [8]. More recently, a mouse model that carries the 3.0 Mb deletion, Del(3.0Mb), has been generated for the first time [9]. Del(3.0Mb)/+ mice were extensively examined for their behavioral phenotypes and were found to exhibit reduced prepulse inhibition, which is known to be an important symptom of schizophrenia, and memory impairment as assessed using a fear-conditioning test [9]; both phenotypes were commonly observed with the 1.5 Mb deletion models [8]. On the other hand, there are some behavioral features only seen in Del(3.0Mb)/+ mice. Del(3.0Mb)/+ mice were reported to be hypoactive in the open field test, while other model mice with the 1.5 Mb deletion are hyperactive, and have a disturbed sleep/wakefulness cycle [9,10]. These problems are generally seen in 22q11.2DS [11,12]. Despite these extensive studies of behavioral phenotypes, the structural changes in the brains are largely unknown. Several human MRI studies have been conducted and revealed overall increased thickness, reduced surface area, and reduced volumes of cortical gray matter in 22q11.2DS, with some exceptions in an area-specific manner [13,14,15,16,17]. Meanwhile, an MRI study of a mouse model carrying the 1.5 Mb deletion has reported no significant changes in the volume of the cerebral cortex compared to the wild type [18]. Although these data provided information on volumetric changes in the brain regions of patients and model mice, alterations in the microscopic structures are not able to be examined by the MRI method. Subcellular brain abnormality was only described in a 22q11.2DS mouse model; it was the reduced spine density of hippocampal pyramidal neurons [19]. However, the cross-areal analyses of the neuron phenotype are missing and therefore the neural circuits involved in the pathogenesis are still largely unknown.

In this study, we performed histological analysis of Del(3.0Mb)/+ mice. We first investigated several anatomical aspects of embryonic and adult brains, but could not detect any changes in overall gross structures. We then analyzed the morphologies of individual neurons in the various brain regions and found that the number of dendritic arborizations and/or spines on them were decreased in the nucleus accumbens (NAc), medial prefrontal cortex (mPFC), and somatosensory area. We further examined axonal tracts of dopaminergic neurons and found that the innervation of dopaminergic axons was reduced in the mPFC. This study provides histological information about the brain abnormalities in Del(3.0Mb)/+ mice, and of the predicted structural basis for psychiatric symptoms of this disease.

## 2. Materials and Methods

### 2.1. Animals

Frozen fertilized eggs of Del(3.0Mb)/+ mouse were obtained from Nagoya University [9], and were transferred to a recipient to generate founder mice at the Institute for Developmental Research. Del(3.0Mb)/+ mice were maintained by crossing heterozygote males with wild type (WT; C57BL/6J) females (both 8–24 weeks). Genotypes were determined by PCR using the primers designed basically on the same site as described by Saito et al. [9] with slight shifting (4–5 base pairs) to improve the amplification efficiency: Common reverse primer designed in *Hira* exon 24 (downstream of the deleted region), 5′-ggccagagagacaagtagagagggagtgg-3′; WT specific forward primer in *Hira* intron 24 (within the deleted region), 5′-gatacggctgctgtgtgtgtgaagcctatg-3′; Del(3.0Mb)-specific forward primer in *Pi4ka* intron 47 (upstream of the deleted region), 5′-ggacccagcaagtccttgcatatttcacgg-3′. All animals were housed at a temperature of 22–24 °C with 40–60% humidity, under a 12 h light/dark cycle (light on at 07:00, off at 19:00), with free access to food and water. Noon on the date that a vaginal plug was observed was considered as embryonic day 0.5 (E0.5). 

### 2.2. Plasmids

Human histone H2B cDNA was amplified using PCR, and inserted upstream of enhanced green fluorescent protein (EGFP) of CAG promoter-driven EGFP expression vector (pCAG-EGFP) [20,21] to generate H2B-EGFP fusion protein expression vector (pCAG-H2B-EGFP). pCAG-turboRFP was generated by exchanging the turboRFP cDNA (a red fluorescent protein, Evrogen, Moscow, Russia) with EGFP of pCAG-EGFP.

### 2.3. In Utero Electroporation

Surgery on pregnant mice and manipulation of embryos in the uterus were performed as previously described [20,22]. Male Del(3.0Mb)/+ and female WT mice were crossed. At E14, the pregnant mice were deeply anesthetized with a mixture of medetomidine (0.3 mg/kg), midazolam (4 mg/kg), and butorphanol (5 mg/kg). In total, 1 µL of plasmid mix composed of pCAG-H2B-EGFP (0.3 mg/mL) and pCAG-turboRFP (0.5 mg/mL) was injected into the lateral ventricle of embryos using a glass micropipette. Then, each embryo was placed between a tweezer-type disc electrode (5 mm in diameter) (CUY650-5; NEPA Gene, Chiba, Japan) and 5 electronic pulses were applied (35 V, 50 ms with intervals of 450 ms) using an electroporator (NEPA21; NEPA Gene). Brains were fixed at E18.

### 2.4. Production of Adeno-Associated Virus

All recombinant DNA experiments were performed in accordance with the protocol approved by the recombinant DNA experiments committee of the Institute for Developmental Research (#21-2). The AAV plasmid carrying EGFP driven by human Synapsin I promoter was developed by modifying AAV-shRNA-ctrl (#85741, Addgene, Watertown, MA, USA). We substituted the region from the U6 promoter until human growth hormone polyadenylation signal (hGH polyA signal) to the synthesized DNA fragment (Biomatik, ON, Canada) which contained U6 promoter, human synapsin I promoter with EGFP, WPRE, and hGH polyA signal. AAV production was performed in accordance with the previous report [23], which is a minor modification of the protocols reported by Challis et al. [24]. HEK293FT cells were passaged and cultured in 10–15 cm diameter dishes in DMEM supplemented with 10% FBS, penicillin, and streptomycin. Cells were transfected with the plasmids pUCmini-iCAP-PHP.eB (Addgene #103005), pAdDeltaF6 (#112867, Addgene), and the plasmid-carrying rAAV genome (pAAV control). To collect AAV particles from cells and culture media, cell pellets were dissociated with 1 mg/mL of DNase I solution (#11284932001, Roche, Mannheim, Germany) and subjected to six freeze–thaw cycles. The AAV-containing solution from cell pellets was collected and placed at 4 °C. The culture medium was mixed with 40% PEG8000, followed by incubation on ice for 2 h. AAV particles were precipitated by centrifugation at 6000× *g*, 4 °C for 30 min, and dissociated with DNase I solution. The AAV-containing solution from cell pellets and the culture medium were mixed and ultracentrifuged at 160,000× *g* with Opti Prep gradients. Purified AAV was concentrated and buffer-exchanged to PBS (−). The virus was titrated using THUNDERBIRD SYBR qPCR Mix (Toyobo, Osaka, Japan) and a CFX96 real-time PCR system (Bio-Rad Laboratories, Hercules, CA, USA), with the following primer sets targeting the WPRE sequence: GGCTGTTGGGCACTGACAAT and CCGAAGGGACGTAGCAGAAG.

### 2.5. Intravenous Administration of Adeno-Associated Virus

Mice were anesthetized under 5% isoflurane until the loss of sensation was ensured in the chamber box. Mice were moved to the warm pad and the virus administration was performed via retro orbital route under 2–3% isoflurane [25]. In total, 30–50 µL of 1 × 10^11^ vg/mL of virus solution was injected into each animal.

### 2.6. Immunofluorescence

For sampling the E18 brains, the pregnant mice were deeply anesthetized with isoflurane, and killed by cervical dislocation. The uteruses were dissected out and kept in ice-cold PBS. The individual embryos were transcardially perfused with 4% paraformaldehyde (PFA) in phosphate-buffered saline (PBS). For sampling the postnatal brains, mice were deeply anesthetized with isoflurane, and perfused with PBS followed by 4% PFA in PBS. After perfusion, the embryonic and adult brains were dissected out and soaked in 4% PFA for at least 2 h. After washing with PBS, the brains were sectioned coronally at 100 μm thickness using a vibrating microtome (VT1000, Leica Biosystems, Wetzlar, Germany). Brain slices obtained from around the level of the interventricular foramen were used. For immunostaining of embryonic brains, sections were placed onto MAS-coat slides (Matsunami Glass, Osaka, Japan) and treated with HistoVT One (Nacalai Tesque Inc., Kyoto, Japan, Cat# 06380-05) at 70 °C for 20 min. After washing with PBS containing 0.05% Tween (PBST), the sections were blocked with 4% BSA in PBST and treated with primary antibodies. The primary antibodies used in this study were anti-GFP (1:3000, Aves Labs, Tigard, OR, USA, Cat# GFP-1010, RRID: AB_2307313, chicken IgY), anti-TAG1 (1:100, R&D Systems, Minneapolis, MN, USA, Cat# AF4439, RRID: AB_2044647, goat IgG), anti-Ctip2 (1:500, Abcam, Cambridge, UK, Cat# ab18465, RRID: AB_2064130, rat IgG), anti-L1 (1:1000, Millipore, Billerica, MA, USA, Cat# MAB5272, RRID: AB_2133200, rat IgG), anti-calbindin [26] (1:2000, rabbit IgG), anti-Cux1 (1:200, GeneTex Inc., Irvine, CA, USA, Cat# GTX56275, rabbit IgG), and anti-tyrosine hydroxylase (1:2000, Novus Biologicals, Littleton, CO, USA, Cat# NB300-109, RRID: AB_10077691, rabbit IgG). After washing with PBST three times, the sections were stained with secondary antibodies: Alexa Fluor 488-conjugated anti-chicken IgY (1:1000, donkey, Jackson ImmunoResearch Laboratories, West Grove, PA, USA, Cat# 703-545-155), Alexa Fluor 488-conjugated anti-rabbit IgG (1:1000, donkey, Jackson ImmunoResearch Laboratories, Cat# 711-545-152), Alexa Fluor 555-conjugated anti-rabbit IgG (1:1000, donkey, Invitrogen, Carlsbad, CA, USA, Cat# A31572), and Alexa Fluor 647-conjugated anti-goat IgG (1:1000, donkey, Jackson ImmunoResearch Laboratories, Cat# 705-605-147). In the cases of staining with rat antibodies (anti-Ctip2 and anti-L1), sections were incubated with biotin-conjugated anti-rat IgG (1:1000, donkey, Jackson ImmunoResearch Laboratories, Cat# 712-065-153), and then incubated with Alexa Fluor 555-conjugated streptavidin (1:1000, Invitrogen, Cat# S32355). Next, 4′,6-diamidino-2-phenylindole (DAPI; 0.2 μg/mL, Sigma-Aldrich St. Louis, MO, USA, Cat# D9542) was used for staining DNA. For adult brains, floating sections were incubated in Epitope Retrieval Solution pH 9 (Leica Biosystems, Cat# RE7119-CE) at 50 °C for 3 h, and processed in the same component of blocking and antibody reactions for the embryonic brains m. The floating sections were then placed on glass slides. Fluorescent images were captured with a confocal laser microscope (LSM880; Carl Zeiss, Oberkochen, Germany).

### 2.7. Golgi–Cox Staining and Spine Analysis

Golgi–Cox staining was performed using the FD Rapid GolgiStain kit (FD NeuroTechnologies, Columbia, MD, USA) following the product instructions with slight modifications. WT (C57BL/6J) and Del(3.0Mb)/+ mice (6–12 months old, male) were deeply anesthetized with isoflurane and decapitated. The brains were quickly harvested, rinsed with double distilled water (DDW), and immersed in a 1:1 mixture of FD Solution A:B for 2 weeks at room temperature in the dark. Brains were then transferred to FD Solution C or tissue-protectant solution (20% Sucrose, 15% glycerol in DDW) [27] and kept in the dark at 4 °C for 48 h. The brains were sectioned coronally in the tissue-protectant solution at 100 μm thickness using a vibrating microtome (VT1000, Leica Biosystems). The sections were mounted on gelatin-coated slides and stained as described in the instructions. After being dehydrated, the slides were mounted with Permount (Fisher Scientific, Pittsburgh, PA, USA). Next, 20–30 Z-stack images with 1 μm intervals of Golgi-stained dendrites were taken using a 100× lens with 1.5× digital zoom on a bright field microscope (BZ-9000, Keyence, Osaka, Japan). The images were processed with StackReg (http://bigwww.epfl.ch/thevenaz/stackreg/, accessed on 27 April 2023) and Stack Focuser (https://imagej.nih.gov/ij/plugins/stack-focuser.html, accessed on 27 April 2023), plugins of Fiji (http://fiji.sc, accessed on 27 April 2023). The number of spines on each dendrite within a 50–70 μm stretch 20–30 μm away from the cell soma were counted using Dendritic Spine Counter (https://imagej.net/plugins/dendritic-spine-counter, accessed on 27 April 2023), a plugin of Fiji.

### 2.8. Dendritic Arbor and Spine Density Analyses of GFP-Labeled Neurons

After immunohistochemistry of AAV-GFP-labeled brains as described above, fluorescent images were obtained using a confocal laser scanning microscope (LSM880, Carl Zeiss). For analysis of the dendritic arbor, 40–60 Z-stack confocal images with 0.79 μm intervals were obtained using a 20× lens with 1.5× digital zoom. Dendrites were traced with Simple Neurite Tracer (SNT, https://github.com/morphonets/SNT/, accessed on 27 April 2023), a plugin of Fiji, and evaluated by Sholl analysis in the same plugin. For spine density analysis, 40–60 Z-stack confocal images with 0.3 μm intervals were obtained using a 63× lens with 2.5× digital zoom. The number of spines on each dendrite within a 20–50 μm stretch located 20–30 μm away from the cell soma were analyzed using Filament Tracer function of Imaris software (version 9.2.0, Bitplane, Zurich, Switzerland).

### 2.9. Statistical Analysis

For all cell imaging experiments, counting and traces were assessed in a blinded manner. Comparisons between two groups were performed using two-sided unpaired Welch’s *t*-test using Prism 7 (GraphPad Software, San Diego, CA, USA). Statistical significance for Figure 1C (bins and genotypes) and Figure 4B–E (distance from soma and genotypes) was determined using two-way ANOVA using Prism 7. *p* < 0.05 (WT vs. Del(3.0Mb)/+) was considered statistically significant.

## 3. Results

### 3.1. Cortical Excitatory Neurons Migrate Normally in Embryonic Del(3.0Mb) Heterozygous Mice

The genome region deleted in Del(3.0Mb) mice includes more than 40 genes, most of which are expressed in the embryonic brain [28], raising the possibility that some aspects of brain development may be altered. During brain development, cortical excitatory neurons migrate from the ventricular zone toward the brain surface. This process is frequently affected by blocking the expression of genes involved in the brain development [29]. We, therefore, examined this possibility in embryonic Del(3.0Mb)/+ mice. Del(3.0Mb)/+ and WT mice were crossed and the resulting embryos were electroporated in utero with pCAG-H2B-EGFP (0.3 mg/mL) together with pCAG-turboRFP (0.5 mg/mL) at E14 (Figure 1A). The embryos were fixed 4 days later (E18), and we examined the distribution of cells labeled with EGFP (nuclei) and their individual cell morphology visualized by turboRFP (cytosol). The labeled cells in WT embryos mostly migrated to the superficial layer of the cortical plate (bin1), and those in Del(3.0Mb)/+ embryos in the same litter also reached the same position (Figure 1B,C). The morphologies of migrating neurons with a thick leading process pointing to the brain surface were not affected in Del(3.0Mb)/+. Hence, the radial migration of cortical excitatory neurons is thought to be normal in Del(3.0Mb)/+ mice.

### 3.2. Overall Histological Organization of Embryonic Del(3.0Mb) Heterozygous Mice Is Normal

We next examined the layer structure of embryonic cortical plates, which can be disturbed by altered neurogenesis, migration, and apoptosis. To assess this, we stained the coronal sections of E18 WT and heterozygous brains with Ctip2, a layer V and VI marker [30], and TGA1, a maker for cortico-fugal and cortical commissure axons [31] (Figure 2A). We measured the thickness of 3 sectors; from layer II to IV (above the Ctip2 positive sector), layers V and VI (Ctip2 positive sector), and from the IZ to the VZ (from the TAG1 positive area to the ventricular surface) in the somatosensory area. However, we could not detect significant changes between the two genotypes in any sectors (Figure 2B). We also examined immunohistochemistry of TAG1 and L1, a pan-axonal marker [32], in coronal sections and found that there was no gross difference in the axonal tracts in the E18 brains (Figure 2C). The thickness of the corpus callosum was not significantly changed between WT and Del(3.0Mb)/+ (Figure 2D). A previous study reported that the migration of interneurons from the medial ganglionic eminence to the cerebral cortex was impaired in E18 Df1/+ mice, which is another 22q11.2DS model carrying a 1.5 Mb deletion [33]. We therefore evaluated the density of interneurons using an early marker, calbindin [26,34], in the Del(3.0Mb)/+ embryonic cerebral cortex at E18; however, there was no significant change compared to WT (Figure 2E,F). These observations indicate that there is no noticeable defect in the cortical development in Del(3.0Mb)/+ mice.

### 3.3. Gross Layer Structure of Adult Del(3.0Mb)/+ Is Preserved

After birth, the dendrite arborization, axon extension, synapse formation, and myelination proceed, which establish the mature layer structures of the cerebral cortex in an area-specific manner. We examined this in adult heterozygous mice. We immunostained the adult brains (6 months) with Cux1, a marker for layer II to IV neurons [35] (and scattered staining can be observed in the lateral part of layer VI), and Ctip2, which is expressed both in layer V and layer VI neurons (Figure 3A). However, the layer V neurons expressed higher Ctip2 and their cell size was larger than layer VI neurons, which enabled us to distinguish them. Based on these features, we identified layers II~IV, V, VI, and white matter in the somatosensory area and measured their thickness (Figure 3B). As a result, we did not detect any changes between WT and heterozygote mice at the level of gross histology.

### 3.4. Dendritic Arbors of Subsets of Neurons in Adult Del(3.0Mb)/+ Are Slightly but Significantly Reduced

Next, we investigated the individual cell morphology of neurons. To this end, we used an adeno-associated virus (AAV) vector expressing EGFP under the control of a neuron-specific synapsin promoter. Virus particles in low titer (10^11^ virus genome/mL) were injected intravenously, and the brains were sampled 7 days later (Figure 4A). We assessed the dendritic branch of GFP-labeled layer V neurons in the somatosensory area (Figure 4B) and in the medial prefrontal cortex (mPFC, Figure 4C), because layer V neurons were preferentially labeled in this method. We found that the dendritic branch of neurons in the mPFC was significantly reduced as assessed by Sholl analysis (the number of intersections at each radius was plotted as a function of distance from the cell body). 

22q11.2 deletion is a high risk factor particularly for schizophrenia and early-onset Parkinson’s disease [36]. Both diseases can be caused by abnormalities in the dopamine system. The axonal targets of dopamine neurons from the ventral midbrain are the striatum, nucleus accumbens (NAc), and mPFC. We therefore investigated the morphology of neurons in the striatum and NAc. As a result, dendritic branch numbers of medium spiny neurons, but not of striatum neurons, in Del(3.0Mb)/+ NAc were significantly decreased (Figure 4D,E). These data suggest that a subset of neurons involved in the dopamine system are selectively affected in Del(3.0Mb)/+ mice.

### 3.5. Dendritic Spine Densities of Subsets of Neurons in Del(3.0Mb)/+ Are Reduced

An anomaly in dendritic spine formation is one of the key features of neurodevelopmental disorders. We next focused on this point, conducted a computer-assisted count of dendritic spines of AAV-labeled neurons. Consequently, the spine density of layer V pyramidal neurons was significantly reduced compared to WT in the somatosensory area of Del(3.0Mb)/+ mice, while there were no significant changes in medium spiny neurons in the striatum and NAc (Appendix A). As an alternative method, we evaluated the spine densities using the Golgi–Cox staining method. In accordance with the results obtained from AAV-labeled neurons, the spine density of layer V neurons in the somatosensory area was reduced, and those of medium spiny neurons in the striatum and NAc were not significantly changed (Figure 5A–C). Additionally, we also found that the spine densities of layer II/III neurons in the somatosensory area and layer V neurons in the mPFC were reduced (Figure 5D,E). We could not detect a significant difference in the spine density of CA1 neurons (the first branch of apical dendrites) between Del(3.0Mb)/+ and WT (Figure 5F). This seems controversial to the previous study showing a slight decrease in LgDel mice, which is another 1.5 Mb deletion model [19]. That study demonstrated that the instability of spines increased, and both formation and elimination of spines increased in that mouse model, resulting in decreased spine density as a net movement. There is another study showing the instability of spines in Df(16)A+/−, another 1.5 Mb deletion model, and it showed unchanged spine density of CA1 neurons in the fixed brain [37]. Therefore, our observation is not necessarily contradictory to the previous studies. Taken together, the spine densities of pyramidal neurons in the somatosensory and mPFC were reduced, while those in basal forebrains and hippocampus were not.

### 3.6. Axon Innervation of Dopaminergic Neurons into the mPFC in Del(3.0Mb)/+ Are Reduced

We observed that the dendrite branches and/or dendritic spines in the mPFC and NAc were reduced. As we mentioned above, these two brain regions are major targets of axons coming from dopaminergic neurons located in the ventral tegmental area (VTA), and they form the dopamine system, whose disruptions are known to be related to some symptoms of schizophrenia and other psychiatric diseases [38]. We, therefore, asked whether this system is affected in Del(3.0Mb)/+ mice. The overall axonal tract of dopaminergic neurons, however, appeared to be normal when tyrosine hydroxylase (TH) was labeled by immunohistochemistry (Figure 6A). On the other hand, within the mPFC, the innervation of TH axons was observed to be fewer in Del(3.0Mb)/+ than in WT (Figure 6B,C), suggesting that axonal projection from VTA to the mPFC (mesocortical pathway) is reduced in Del(3.0Mb)/+ mice. 

## 4. Discussion

The microdeletion at the 22q11.2 locus is associated with genetic risk for various psychiatric diseases including schizophrenia. Here, we conducted histological analyses of a 22q11.2DS model mouse, Del(3.0Mb)/+, which mimics the 3.0 Mb deletion carried by the most (90%) 22q11.2DS patients. We first analyzed the embryonic brains of Del(3.0Mb)/+ mice at E18, but could not detect changes in the migration of cortical excitatory neurons, layer structure, axon bundles, or distribution of interneurons in the cerebral cortex. Subsequent investigation of the adult brain again found no gross difference in the layer structure of the cerebral gray and white matter. We thus analyzed the morphologies of individual neurons in the various brain regions, and found that dendritic arborizations and spines on them were decreased in several regions, i.e., the NAc, mPFC, and cortical somatosensory area. Because the NAc and mPFC are known to be the main axonal targets of dopaminergic neurons, we examined their axonal tracts, and found that the innervation of dopaminergic axons was reduced in the mPFC, indicating the deficits in dopamine system in this model animal. This study is the first attempt to perform comprehensive histological analyses of the 22q11.2DS mouse model brains, and provides basic information on the structural basis for psychiatric symptoms of this disease.

Previous studies demonstrated that several genes located in the deletion region were extensively expressed during the mouse brain development [28], suggesting that their functional deficiency is involved in neurodevelopmental phenotypes. However, no abnormality was observed in the embryonic brains of Del(3.0Mb)/+. In many cases, 22q11.2DS patients do not have structural brain malformations such as band heterotopia and gyrification abnormalities, supporting our proposal that the hemizygosity of these genes does not affect the gross development of the brain [14]. On the other hand, several MRI studies revealed volumetric reductions in the cortical gray and white matter, and subcortical structures in individuals with 22q11.2DS [13,14,15,16,17,39], whereas we could not detect such changes in the somatosensory area of adult Del(3.0Mb)/+ similar to the 1.5 Mb deletion model [18]. The human cerebral cortex might be more severely affected than that of mice by the deletion of 22q11.2 or its syntenic region. Another possibility is that the individuals tested are those selected to be less severe, because about 70% of Del(3.0Mb)/+ mice die neonatally [9]. During brain development, GABAergic interneurons migrate from medial ganglionic eminence to the cortical hemisphere. Toritsuka et al. have shown that a model of 22q11.2DS, Df1/+ mice, has deficits in the migration of interneurons at E18 [33]. In this study, we observed that the density of calbindin^+^ interneurons in the Del(3.0Mb)/+ cerebral hemisphere was normal at E18. As Df1/+ mice carry a 1.5 Mb deletion, this discrepancy might be due to the difference in the deleted region or the difference in interneuron markers used. Further studies are needed to elucidate this point.

In the adult brain of Del(3.0Mb)/+, we observed the deficits in the dopamine system, including the reduced dendritic branches and/or spine densities of pyramidal neurons in NAc and mPFC, as well as the reduced innervation of dopaminergic axons in the mPFC. These observations imply that the dopamine system may be attenuated in Del(3.0Mb)/+ mice. Several lines of evidence support this possibility. (1) Perturbation of the dopamine system is considered to be related to pathologies of schizophrenia and other psychiatric diseases whose risks are increased in 22q11.2DS [38]. (2) Del(3.0Mb)/+ mice exhibited lower prepulse inhibition (PPI) [9], which is probably reflecting the fact that patients of schizophrenia, as well as healthy volunteers applied with a D2 dopamine receptor agonist, showed reduced PPI [40,41,42]. (3) The 22q11.2 deletion also increases the risk of early-onset Parkinson’s disease [6], which is caused by disruption of dopamine system [43]. (4) Dopaminergic neurons differentiated from induced pluripotent stem cells (iPSC) derived from patients of 22q11.2DS, which showed impaired neurite outgrowth [44]. (5) The *COMT* gene included in the 22q11.2 deleted region [9] encodes catechol-methyltransferase (COMT), which degrades catecholamines including dopamine released in the synaptic cleft, especially in the prefrontal cortex [45]. One of the genetic variants of *COMT*, Val158Met (rs4680), which decreases enzymatic activity by 40% compared to the WT, is thought to be one of the risks for schizophrenia [46]. Based on all these reports and knowledge, and our data provided here, the dopamine system is supposed to be disturbed in Del(3.0Mb)/+ mice. Other than the dopamine system, we also observed reduced spine density both in layer II/III and in layer V in the cortical somatosensory area. This might indicate the abnormal sensory input of Del(3.0Mb)/+ mice, or more likely, the spine density may be reduced globally across the neocortex. A previous diffusion MRI study suggested reduced axon trajectory in the superior longitudinal fasciculus in 22q11.2DS, indicating weakened interregional connectivity [47]. This might explain some behavioral abnormalities observed in this mouse model, such as smaller field potential changes evoked by visual stimuli in the primary visual cortex [9].

Our histological analyses of Del(3.0Mb)/+ revealed that dendritic branches and/or spine density were reduced in the specific brain regions, whereas gross brain development was normal. When and how these changes develop should be determined in future studies.

## 5. Conclusions

In the present study, we conducted comprehensive histological analyses of the embryonic and adult Del(3.0Mb)/+ mouse brains. No gross anatomical changes were found in the embryonic stage, including the migration of cortical excitatory neurons, cortical layer and axon formation, and density of cortical interneurons. Instead, dendritic branches and/or dendritic spine density were reduced in the NAc, mPFC, and somatosensory area in the adult stages. These observations indicate that there are histological changes at least in the brain regions related to the dopamine system in Del(3.0Mb)/+ mice.

## Figures and Tables

**Figure 1 biomolecules-13-00763-f001:**
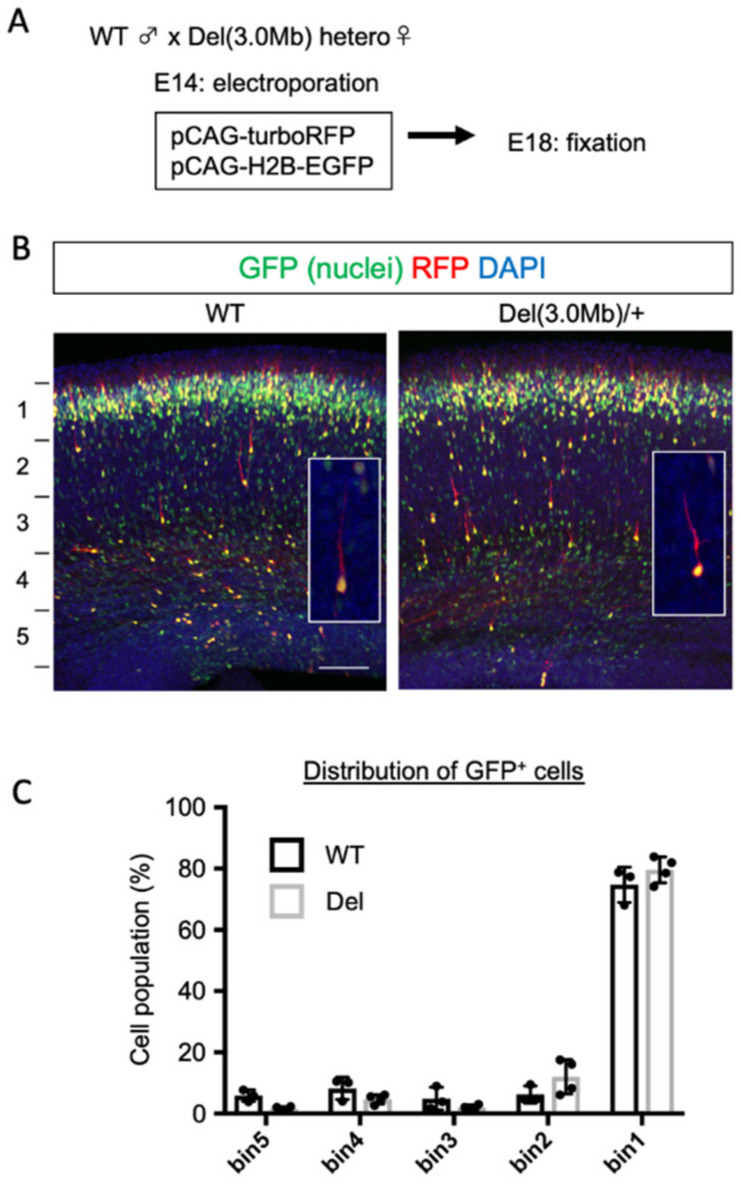
The radial migration of cortical excitatory neurons in Del(3.0Mb)/+ embryos was indistinguishable from those of WT. (**A**) Experimental design of in utero electroporation. The plasmid mix containing pCAG-turboRFP (0.5 μg/μL) and pCAG-H2B-EGFP (0.3 μg/μL) was electroporated into the littermates that resulted from crossing between a male Del(3.0Mb)/+ and a female wild type (WT) mouse at embryonic day 14 (E14), and then fixed at E18. (**B**) Electroporated neurons of Del(3.0Mb)/+ embryos migrated mostly into the top of the cortical plate, which was not indistinguishable from those of WT. The slices of electroporated brains were stained with DAPI (nuclei, blue). The fluorescent signals of turboRFP (red, cytosol) and EGFP (green, nuclei) were merged. High magnification images of individual migrating neurons are shown in the inlets. Scale bar: 100 μm. (**C**) Histogram of GFP^+^ cell position in the cortical thickness. The thickness from the border between intermediate zone and ventricular zone to the surface of the brain was divided into 5 bins. The percentages of the GFP^+^ nuclei in each bin were evaluated. The bars represent the mean ± SD. Two-way ANOVA was conducted (WT: 673 cells from 3 brains, Del: 1126 cells from 4 brains). Significance between the two genotypes was not detected (WT vs. Del, bin1-5, *p* > 0.9999).

**Figure 2 biomolecules-13-00763-f002:**
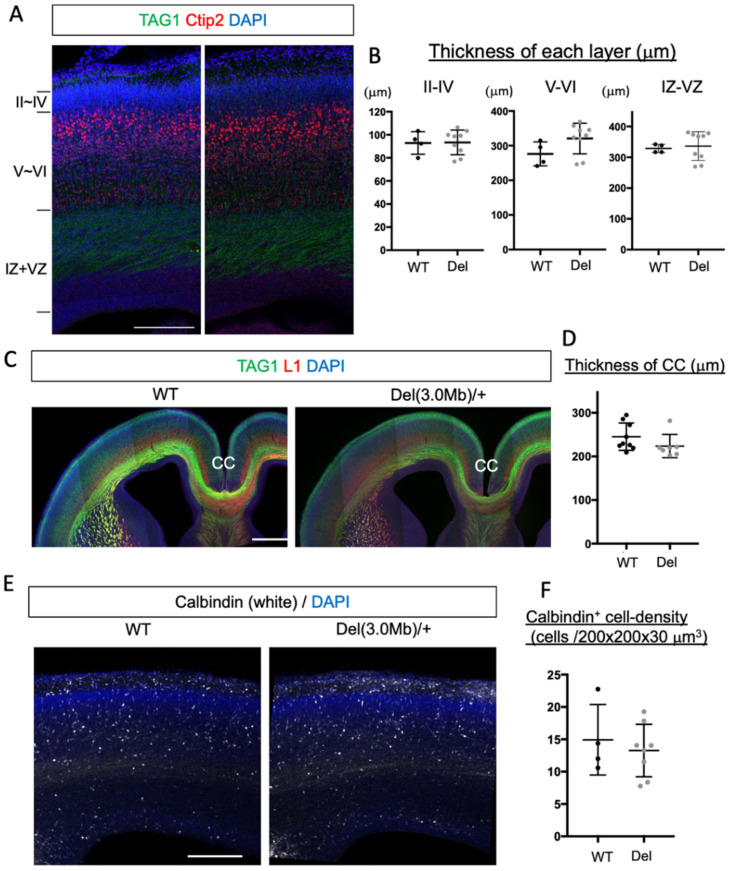
Gross histological organization of embryonic Del(3.0Mb)/+ mice is normal. (**A**) Layer structure of the E18 cerebral cortex. Coronal sections at the somatosensory area were stained with TAG1 (green), Ctip2 (red), and DAPI (blue). Scale bar: 200 μm. (**B**) Quantification of the thickness of layer II–IV (from the upper surface of Ctip2-labeled sector to the upper surface of DAPI dense sector), layer V–VI (Ctip2-labeled sector), and IZ–VZ (from the border between Ctip2 and TAG1 labeled sectors to the ventricular surface of the pallium). The thickness of each sector was not significantly changed in Del(3.0Mb)/+ (2-sided unpaired *t*-test with Welch’s correction, layer II–IV; *p* = 0.9364, layer V–VI; *p* = 0.0881, IZ–VZ; *p* = 0.6593, WT: 4 brains, Del: 9 brains). IZ, intermediate zone; VZ, ventricular zone. (**C**) No gross differences in the axonal tracts in the E18 forebrains were found. Sections of indicated genotypes at the level of somatosensory area were stained with anti-TAG1 (green) and anti-L1 (red) antibodies, and DAPI (blue). CC, corpus callosum. Scale bar: 500 μm. (**D**) Quantification of the thickness of CC. The thickness of CC at the midline was not significantly altered between WT and Del(3.0Mb)/+ (2-sided unpaired t-test with Welch’s correction, WT vs. Del; *p* = 0.1588, WT: 9 brains, Del: 7 brains). (**E**) Density of calbindin^+^ interneurons in the cerebral cortex at E18. Sections of indicated genotypes at the level of somatosensory area were stained with anti-calbindin antibody (white), and DAPI (blue). Scale bar: 200 μm. (**F**) No significant differences were detected between the 2 genotypes (2-sided unpaired *t*-test with Welch’s correction, WT vs. Del; *p* = 0.6134, WT: 4 brains, Del: 8 brains). Horizontal bars in scatter plots (**B**,**D**,**F**) represent mean ± SD.

**Figure 3 biomolecules-13-00763-f003:**
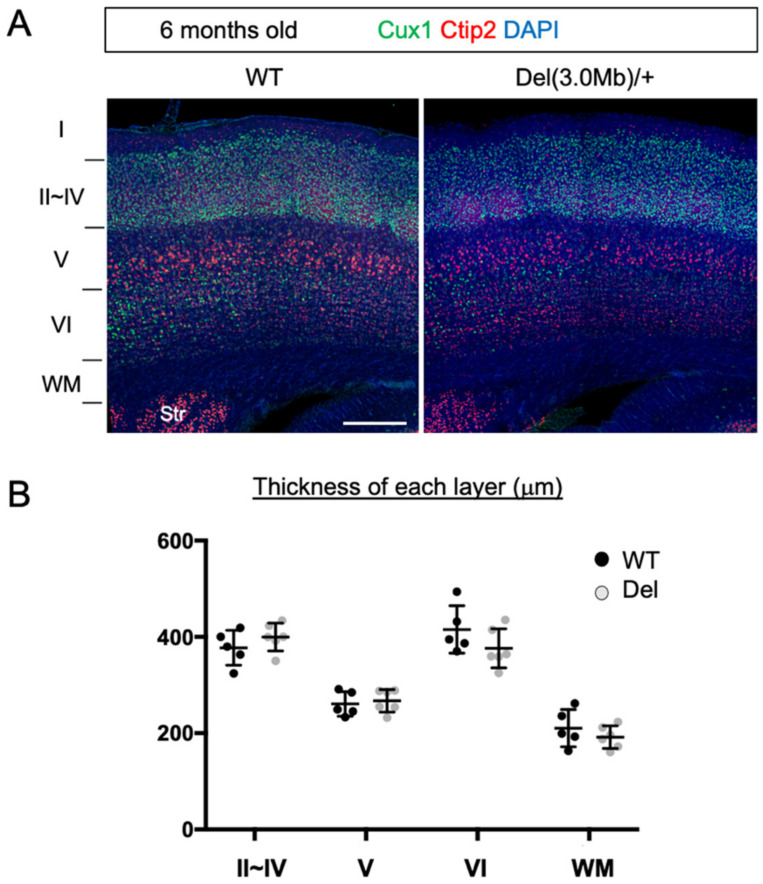
Gross cortical layer structure of adult Del(3.0Mb)/+ is preserved. (**A**) Coronal sections of the adult (6–12 months old) brains at the somatosensory area were stained with anti-Cux1 (green) and anti-Ctip2 (red) antibodies, and DAPI (blue). Scale bar: 300 μm. (**B**) Quantification of the thickness of layer II–IV (Cux1 heavily labeled sector), layer V (Ctip2 heavily labeled sector), layer VI (Ctip2 weakly labeled sector), and the white matter (WM, below layer VI). No significant changes in the thickness of any cortical layers and the WM (2-sided unpaired *t*-test with Welch’s correction, layer II–IV: *p* = 0.2959, layer V: *p* = 0.6645, layer VI: *p* = 0.1943, WM: *p* = 0.3813, WT: 5 brains, Del: 6 brains). WM, white matter. Horizontal bars in scatter plots (**B**) represent mean ± SD.

**Figure 4 biomolecules-13-00763-f004:**
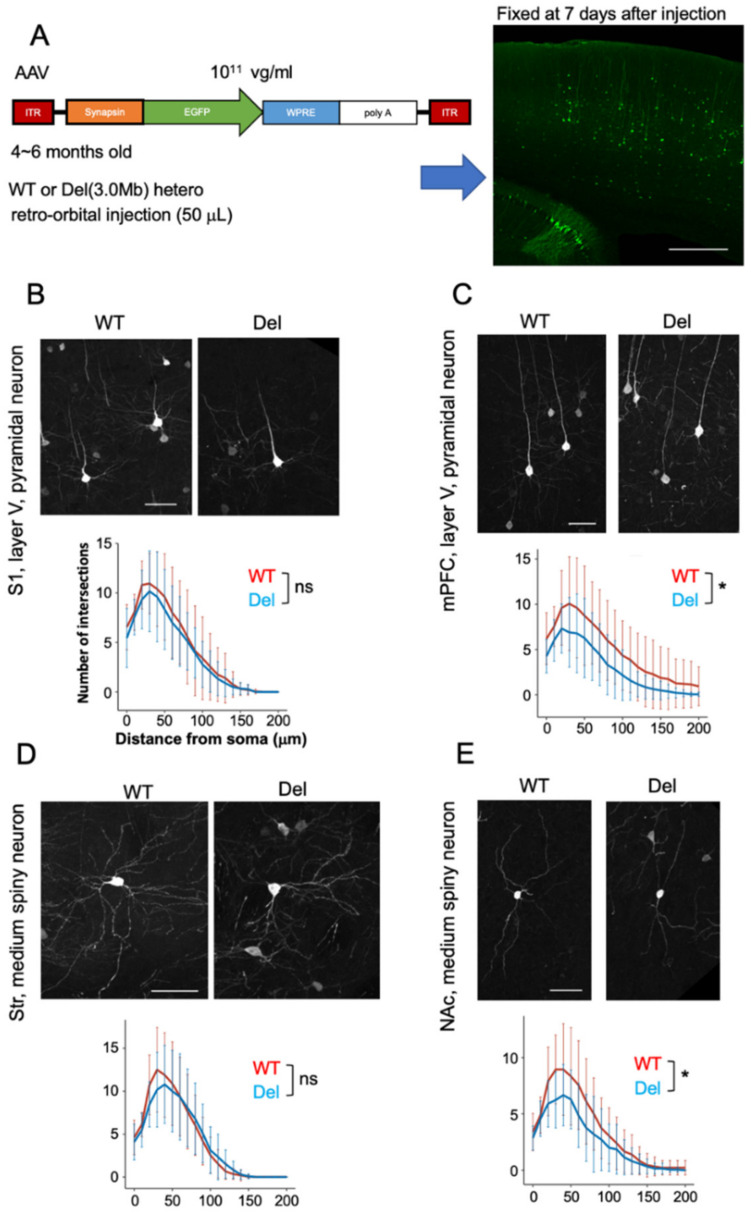
Dendritic arbors of subsets of neurons in adult Del(3.0Mb)/+ are slightly but significantly reduced. (**A**) Experimental design for visualization of individual neurons using EGFP-expressing AAV vector. About 50 µL of AAV solution in low titer (10^11^ virus genome (vg)/mL) was administered to adult (6 months old) mice via the retro-orbital route. These mice were fixed 7 days after the injection. Representative image of a sagittal section of the WT brain was shown (right). Scale bar: 500 μm. (**B**–**E**) Representative images of GFP^+^ individual neurons in various brain regions of WT or Del(3.0Mb)/+. Scale bars: 50 μm. The complexity of dendrite branches was quantified with Sholl analysis (bottom of each panel). Error bars indicate ± SD. The red lines and blue lines in the graphs represent WT and Del(3.0Mb)/+, respectively. The number of intersections in a range of 20–100 μm in the WT and Del(3.0Mb)/+ mouse brains were analyzed with two-way ANOVA (distance from soma and genotypes). The dendrite branches were significantly reduced in Del(3.0Mb)/+ medial prefrontal cortex (mPFC) (**C**) and nucleus accumbens (NAc) (**E**), while no significant changes were observed in the somatosensory area (S1) (**B**) and striatum (Str) (**D**). (**B**) WT: 20 neurons from 4 brains, Del: 18 neurons from 4 brains, F (1, 36) = 0.6035, *p* = 0.4423. (**C**) WT: 15 neurons from 3 brains, Del: 19 neurons from 5 brains, F (1, 32) = 4.585, *p* = 0.0400. * *p* < 0.05. (**D**) WT: 26 neurons from 3 brains, Del: 24 neurons from 5 brains, F (1, 48) = 0.2107, *p* = 0.6483. (**E**) WT: 18 neurons from 4 brains, Del: 21 neurons from 5 brains, F (1, 37) = 5.811, *p* = 0.0210. * *p* < 0.05. *p* values are for comparisons between WT vs. Del(3.0Mb/+). The interaction effects were not significant in (**C**,**E**).

**Figure 5 biomolecules-13-00763-f005:**
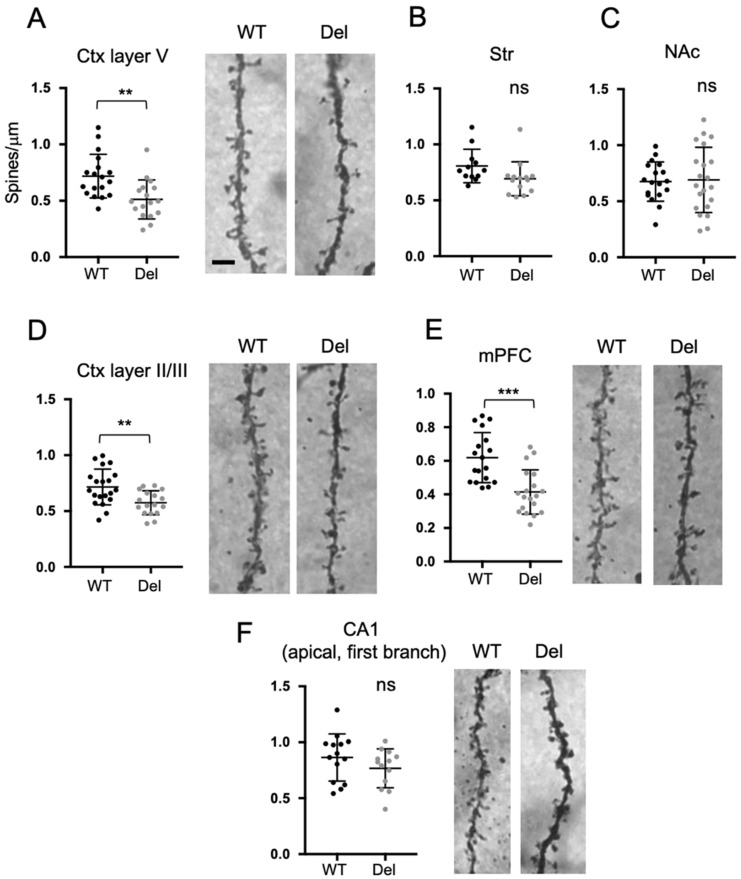
Dendritic spine densities of subsets of neurons in Del(3.0Mb)/+ are reduced. Dendritic spine density was evaluated on Golgi–Cox impregnated brain slices of adult (12 months old) WT or Del(3.0Mb)/+ mice. Representative images of spines visualized with Golgi–Cox staining are shown on the right (**A**,**D**–**F**). In accordance with the results of AAV-labeled neurons, the spine density (spine number/μm) of basal dendrites of pyramidal neurons in layer V in the somatosensory area (**A**) was significantly reduced, while those of medial spiny neurons in the Str (**B**) and in NAc (**C**) were not. In addition, the spine densities of basal dendrites of pyramidal neurons in layer II/III of somatosensory area (**D**) and those in layer V of mPFC (**E**) were reduced, while the significance was not detected in the first branch of apical dendrite of CA1 pyramidal neurons in the hippocampus. Two-sided unpaired *t*-tests with Welch’s correction were conducted. (**A**) *p* = 0.0022, ** *p* < 0.01, WT: 18 dendrites from 3 brains, Del: 17 dendrites from 3 brains. (**B**) *p* = 0.0665, WT: 12 dendrites from 3 brains, Del: 14 dendrites from 3 brains. (**C**) *p* = 0.8544, WT: 18 dendrites from 3 brains, Del: 21 dendrites from 3 brains. (**D**) *p* = 0.0030, ** *p* < 0.01, WT: 20 dendrites from 4 brains, Del: 17 dendrites from 4 brains. (**E**) *p* = 0.0001, *** *p* < 0.001, WT: 18 dendrites from 4 brains, Del: 19 dendrites from 4 brains. (**F**) *p* = 0.2015, WT: 15 dendrites from 3 brains, Del: 16 dendrites from 3 brains. ns, not statistically significant. Scale bars: 3 μm. Horizontal bars in scatter plots represent mean ± SD.

**Figure 6 biomolecules-13-00763-f006:**
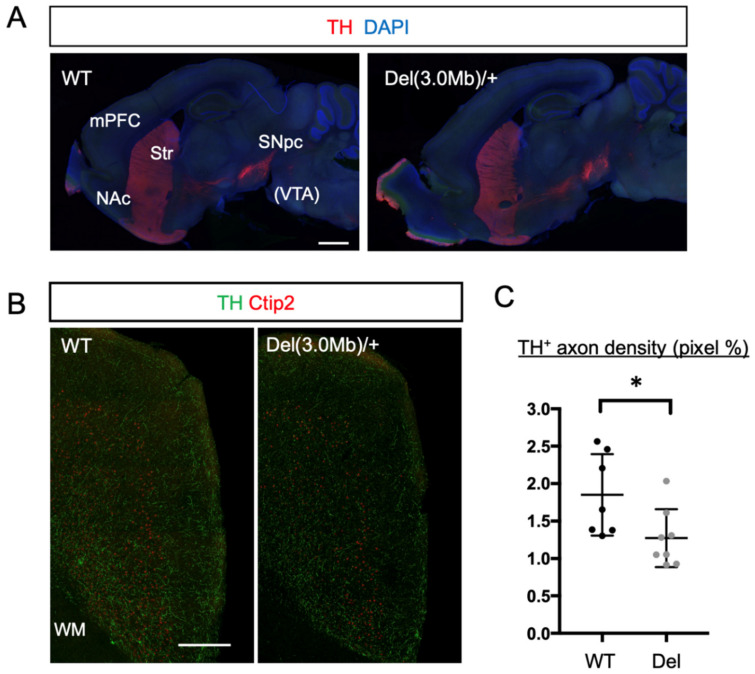
Axon innervation of dopaminergic neurons into the mPFC in Del(3.0Mb)/+ is reduced. (**A**) Gross axonal tracts of dopaminergic neurons were unaffected. Sagittal sections of WT or Del(3.0Mb)/+ mouse brains were stained with anti-tyrosine hydroxylase (TH, red) antibody and DAPI (blue). SNpc, substantia nigra pars compacta. VTA, ventral tegmental area. VTA is located in different sagittal planes. Scale bar: 1 mm. (**B**) TH axons in the prefrontal area are reduced in Del(3.0Mb)/+ mice. Coronal sections of prefrontal area were stained with anti-TH (green) and anti-Ctip2 (red) antibodies. The right margin of the tissue section corresponds to the midline of the brain. Scale bar: 300 μm. (**C**) Quantification of the density of TH axons. Two-sided unpaired *t*-test with Welch’s correction, WT vs. Del; *p* = 0.0397, * *p* < 0.05, WT: 7 brains, Del: 8 brains (6–12 months old). Horizontal bars in scatter plots (**C**) represent mean ± SD.

## Data Availability

The data that support the findings of this study are available from the corresponding authors, upon reasonable request.

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
