# Peer review of "Histological Analysis of a Mouse Model of the 22q11.2 Microdeletion Syndrome"

_biomolecules, 2023, doi:10.3390/biom13050763_

Round 1

Reviewer 1 Report

In this work, Tabata et al. describe the cytoarchitectures of Del(3.0Mb)/+ mouse brains and state that the general histology of the embryonic and adult cerebral cortices was comparable to that of wild-type mice, but the individual neuron morphologies were slightly but significantly changed from their wild-type counterparts in some regions. Furthermore, they observed a reduced innervation of the axons of dopaminergic neurons in the prefrontal cortex, which could explain the anomalous behaviors of Del(3.0Mb)/+ mice.

I scrutinized the manuscript and appreciated the effort made by the authors. The introduction is well written, with adequate bibliographical references. The results are clearly expressed with appropriate graphical representations. The discussion is fair, and relevant to the results obtained. However, some concerns lead me to state that the study needs some revisions. 

Major revisions

-the absence of behavioral tests

The relationship between the deletion and related cognitive symptoms can be challenging to interpret, and it is difficult to establish a cause-and-effect relationship. In addition, behavioral data related to learning, memory, and attention collected in different studies using a 3.0-Mb deletion mimicking model vary considerably. Have you performed behavioral tests on animals to assess which cognitive symptoms were generated by the mutation? In particular, given that in other studies, Del(3.0 Mb)/+ mice have shown impaired early visual processing that is also usually observed in patients with schizophrenia, have you tested this in animals?

-lack of identification of the pathology(s) caused by the deletion.

The deletion used in the study usually increases the risk of also developing other neurological conditions in addition to schizophrenia, such as parkinsonism, autism, and intellectual disability, and this makes it more difficult to relate the results obtained to a specific disease. You state that disruption of the dopamine system causes schizophrenia-related symptoms and that the 22q11.2 deletion increases the risk of early-onset Parkinson's disease, which is caused by the disorder of dopamine. Your results show that there are histological changes, at least in the brain regions related to the dopamine system in Del(3.0Mb)/+ mice. Still, these histological changes cannot be uniquely associated with a single pathology, which could be a limitation for the translational aspect of the study. Try to solve this limitation by defining the pathological phenotype generated by the deletion in mice.

Minor revision

Figure 5 is missing units on the Y axes.

Author Response

Dear Reviewers,

We appreciate the valuable and constructive comments the reviewers raised, which gave us the opportunity to improve our manuscript significantly. We addressed all comments as shown below. All changes in the manuscript were marked up by the “Track Changes” function of MS Word. Figures 1-5 have been also revised according to the reviewer’s comments and have been replaced in the revised manuscript. One of the reviewers claimed that Supplementary Figure 1 was not available. We appreciate it if you could make the link (line 827) active. We hope you will find this revised manuscript suitable for publication in Biomolecules.

<Reviewer1>

In this work, Tabata et al. describe the cytoarchitectures of Del(3.0Mb)/+ mouse brains and state that the general histology of the embryonic and adult cerebral cortices was comparable to that of wild-type mice, but the individual neuron morphologies were slightly but significantly changed from their wild-type counterparts in some regions. Furthermore, they observed a reduced innervation of the axons of dopaminergic neurons in the prefrontal cortex, which could explain the anomalous behaviors of Del(3.0Mb)/+ mice.

I scrutinized the manuscript and appreciated the effort made by the authors. The introduction is well written, with adequate bibliographical references. The results are clearly expressed with appropriate graphical representations. The discussion is fair, and relevant to the results obtained. However, some concerns lead me to state that the study needs some revisions. 

Major revisions

-the absence of behavioral tests

The relationship between the deletion and related cognitive symptoms can be challenging to interpret, and it is difficult to establish a cause-and-effect relationship. In addition, behavioral data related to learning, memory, and attention collected in different studies using a 3.0-Mb deletion mimicking model vary considerably. Have you performed behavioral tests on animals to assess which cognitive symptoms were generated by the mutation? In particular, given that in other studies, Del(3.0 Mb)/+ mice have shown impaired early visual processing that is also usually observed in patients with schizophrenia, have you tested this in animals?

> We thank this valuable comment. 3.0-Mb deletion model, Del(3.0Mb)/+, was actually generated and published recently in 2020. Therefore, the behavioral phenotypes of this model were described only in the first paper published in 2020 (Saito et al). However, the histological analysis of this model had not been examined. We conducted them upon collaboration work with the first developers of this mouse line in Nagoya University. Del(3.0Mb)/+ mice were directly transferred from Nagoya University, so we did not perform behavioral tests on our side. We have revised the introduction (lines 51-56, and line 72) so that this background can be clearly conveyed.

-lack of identification of the pathology(s) caused by the deletion.

The deletion used in the study usually increases the risk of also developing other neurological conditions in addition to schizophrenia, such as parkinsonism, autism, and intellectual disability, and this makes it more difficult to relate the results obtained to a specific disease. You state that disruption of the dopamine system causes schizophrenia-related symptoms and that the 22q11.2 deletion increases the risk of early-onset Parkinson's disease, which is caused by the disorder of dopamine. Your results show that there are histological changes, at least in the brain regions related to the dopamine system in Del(3.0Mb)/+ mice. Still, these histological changes cannot be uniquely associated with a single pathology, which could be a limitation for the translational aspect of the study. Try to solve this limitation by defining the pathological phenotype generated by the deletion in mice.

> This is a very important and interesting point, although we could not determine it in this study. We agree that we cannot determine which behavioral abnormality is associated with which histological abnormality we observed here. We have deleted the description regarding the specific schizophrenia-related symptoms (lines 783-784). The perturbation of the dopaminergic system is known to be related to a broad range of psychiatric diseases. We have revised the sentence as follows: Perturbation of the dopamine system is considered to be related to pathologies of schizophrenia and other psychiatric diseases whose risks are increased in 22q11.2DS. We have also revised the sentences regarding the symptoms of schizophrenia (lines 403, 799).

Minor revision

Figure 5 is missing units on the Y axes.

> Thank you for pointing this out. We have corrected the Y axes.

Reviewer 2 Report

The manuscript entitled “Histological analysis of a mouse model of the 22q11.2 microdeletion syndrome” by Tabata et al. attempts to link – at least partially – some severe psychiatric symptoms with a few selected light microscopic abnormalities in the mouse brain. There are several points that the authors should consider in their revised manuscript.

General remarks:

1) The number of analyzed dendrites for spine densities is way too low. In some instances, there were only 8 apical dendrites analyzed from 3 animals; when a morphological finding is used to explain such diverse and severe psychiatric symptoms (Parkinson’s disease, schizophrenia, autism, etc.), substantially more cytomorphological analysis should be done.

2) There is no experimental proof that the analyzed dendrites were those that received dopaminergic innervation or affected in any way by the dopaminergic system.

3) There are only 8 references (20%) from the past 5 years (6 references are from the 1990s), while most of the cited work is 10-20 years old. The list of references should be refreshed, if possible, and the discussion should refer to those newer findings. Here are some recent references that the authors should consider for inclusion and discuss in the manuscript; in general, these references contradict the finding that the microdeletion has no effect on the brain architecture:

Rogdaki M, Gudbrandsen M, McCutcheon RA, Blackmore CE, Brugger S, Ecker C, Craig MC, Daly E, Murphy DGM, Howes O. Magnitude and heterogeneity of brain structural abnormalities in 22q11.2 deletion syndrome: a meta-analysis. Mol Psychiatry. 2020 Aug;25(8):1704-1717. doi: 10.1038/s41380-019-0638-3. Epub 2020 Jan 10. PMID: 31925327; PMCID: PMC7387301.

Ching CRK, Gutman BA, Sun D, et al. Mapping Subcortical Brain Alterations in 22q11.2 Deletion Syndrome: Effects of Deletion Size and Convergence With Idiopathic Neuropsychiatric Illness. Am J Psychiatry. 2020;177(7):589-600. doi:10.1176/appi.ajp.2019.19060583

Kikinis Z, Cho KIK, Coman IL, et al. Abnormalities in brain white matter in adolescents with 22q11.2 deletion syndrome and psychotic symptoms. Brain Imaging Behav. 2017;11(5):1353-1364. doi:10.1007/s11682-016-9602-x

Padula MC, Schaer M, Armando M, et al. Cortical morphology development in patients with 22q11.2 deletion syndrome at ultra-high risk of psychosis. Psychol Med. 2018;48(14):2375-2383. doi:10.1017/S0033291717003920

Gudbrandsen M, Bletsch A, Mann C, et al. Neuroanatomical underpinnings of autism symptomatology in carriers and non-carriers of the 22q11.2 microdeletion. Mol Autism. 2020;11(1):46. Published 2020 Jun 8. doi:10.1186/s13229-020-00356-z

4) It is curious that neither the mesocortical nor the mesolimbic pathways, presumed to be affected by the 22q11.2 deletion, was mentioned in the text.

5) The microscopic figures should be made larger. They are very good, so why not show them in full bloom? For example, some details in Figure 1B, 2A, 3A, 4A are difficult to see. Similarly, some of the numbers on the graphs are almost impossible to see (Figure 4B, D, and those on the graphs in Figure 5).

Minor points:

6) Lines 76-77: The sentence should be changed to something that indicates that the number dendritic arborizations and/or spines was decreased…

7) Lines 87-90: Reference(s) should be given to explain why these primers were used (location of sequence, etc.)

8) Line 98: What is turboPFG? Please explain this abbreviation.

9) Line 106: mL? Impossible!

10) Line 114: The design and production of adeno-associated virus should be explained in more detail; please use appropriate references for each step.

11) Lines 128, 374: mL? Impossible!

12) Line 132: Embryonal age (E…) should be indicated. How was perfusion of the embryos performed? Please detail it.

13) Line 153: anti-chicken

14) Lines 169-170: What type of “deep anesthesia” was used? It must be explained.

15) Line 185: Fiji must be referenced or linked.

16) Line 256: The rationale to localize calbindin positive neurons is missing. Please provide references for the importance of calbindin positive interneurons.

17) Line 308: the abbreviation EGFP at this fourth appearance is finally explained… Abbreviations should be spelled out when first used.

18) Line 371: Scale to Figure 4A is missing.

19) Lines 375: The use of retro-orbital route should be referenced (for example, Yardeni et al., 2011). This paper states the correct amount of injectable fluid by this route.

Yardeni T, Eckhaus M, Morris HD, Huizing M, Hoogstraten-Miller S. Retro-orbital injections in mice. Lab Anim (NY). 2011 May;40(5):155-60. doi: 10.1038/laban0511-155. PMID: 21508954; PMCID: PMC3158461.

20) Line 398: Correctly: “showing a”

21) Lines 452-458: As state above, the number of analyzed dendrites seems to be too small (for example, 8 dendrites from 3 brains). The number of dendrites analyzed gave a value that was significantly different between wild-type and Del(3.0Mb)/+ animals. Since this observation is one of the main findings of the study, a larger number of analyzed dendrites would have looked much better.

22) Line 543: What is PERK? Why is it important in this study to reference it?

23) Lines 551-553: The authors hypothesize that “This might indicate the abnormal sensory input of Del(3.0 Mb)/+ mice, or more likely, the spine density may be reduced globally across the neocortex.” Are there any literature data to support any of these?

24) Line 569: There was no Supplementary Materials available to this reviewer.

Author Response

Dear Reviewers,

We appreciate the valuable and constructive comments the reviewers raised, which gave us the opportunity to improve our manuscript significantly. We addressed all comments as shown below. All changes in the manuscript were marked up by the “Track Changes” function of MS Word. Figures 1-5 have been also revised according to the reviewer’s comments and have been replaced in the revised manuscript. One of the reviewers claimed that Supplementary Figure 1 was not available. We appreciate it if you could make the link (line 827) active. We hope you will find this revised manuscript suitable for publication in Biomolecules.

<Reviewer1>

In this work, Tabata et al. describe the cytoarchitectures of Del(3.0Mb)/+ mouse brains and state that the general histology of the embryonic and adult cerebral cortices was comparable to that of wild-type mice, but the individual neuron morphologies were slightly but significantly changed from their wild-type counterparts in some regions. Furthermore, they observed a reduced innervation of the axons of dopaminergic neurons in the prefrontal cortex, which could explain the anomalous behaviors of Del(3.0Mb)/+ mice.

I scrutinized the manuscript and appreciated the effort made by the authors. The introduction is well written, with adequate bibliographical references. The results are clearly expressed with appropriate graphical representations. The discussion is fair, and relevant to the results obtained. However, some concerns lead me to state that the study needs some revisions. 

Major revisions

-the absence of behavioral tests

The relationship between the deletion and related cognitive symptoms can be challenging to interpret, and it is difficult to establish a cause-and-effect relationship. In addition, behavioral data related to learning, memory, and attention collected in different studies using a 3.0-Mb deletion mimicking model vary considerably. Have you performed behavioral tests on animals to assess which cognitive symptoms were generated by the mutation? In particular, given that in other studies, Del(3.0 Mb)/+ mice have shown impaired early visual processing that is also usually observed in patients with schizophrenia, have you tested this in animals?

We thank this valuable comment. 3.0-Mb deletion model, Del(3.0Mb)/+, was actually generated and published recently in 2020. Therefore, the behavioral phenotypes of this model were described only in the first paper published in 2020 (Saito et al). However, the histological analysis of this model had not been examined. We conducted them upon collaboration work with the first developers of this mouse line in Nagoya University. Del(3.0Mb)/+ mice were directly transferred from Nagoya University, so we did not perform behavioral tests on our side. We have revised the introduction (lines 51-56, and line 72) so that this background can be clearly conveyed.

-lack of identification of the pathology(s) caused by the deletion.

The deletion used in the study usually increases the risk of also developing other neurological conditions in addition to schizophrenia, such as parkinsonism, autism, and intellectual disability, and this makes it more difficult to relate the results obtained to a specific disease. You state that disruption of the dopamine system causes schizophrenia-related symptoms and that the 22q11.2 deletion increases the risk of early-onset Parkinson's disease, which is caused by the disorder of dopamine. Your results show that there are histological changes, at least in the brain regions related to the dopamine system in Del(3.0Mb)/+ mice. Still, these histological changes cannot be uniquely associated with a single pathology, which could be a limitation for the translational aspect of the study. Try to solve this limitation by defining the pathological phenotype generated by the deletion in mice.

This is a very important and interesting point, although we could not determine it in this study. We agree that we cannot determine which behavioral abnormality is associated with which histological abnormality we observed here. We have deleted the description regarding the specific schizophrenia-related symptoms (lines 783-784). The perturbation of the dopaminergic system is known to be related to a broad range of psychiatric diseases. We have revised the sentence as follows: Perturbation of the dopamine system is considered to be related to pathologies of schizophrenia and other psychiatric diseases whose risks are increased in 22q11.2DS. We have also revised the sentences regarding the symptoms of schizophrenia (lines 403, 799).

Minor revision

Figure 5 is missing units on the Y axes.

Thank you for pointing this out. We have corrected the Y axes.

<Reviewer2>

The manuscript entitled “Histological analysis of a mouse model of the 22q11.2 microdeletion syndrome” by Tabata et al. attempts to link – at least partially – some severe psychiatric symptoms with a few selected light microscopic abnormalities in the mouse brain. There are several points that the authors should consider in their revised manuscript.

General remarks:

1) The number of analyzed dendrites for spine densities is way too low. In some instances, there were only 8 apical dendrites analyzed from 3 animals; when a morphological finding is used to explain such diverse and severe psychiatric symptoms (Parkinson’s disease, schizophrenia, autism, etc.), substantially more cytomorphological analysis should be done.

We have added the analysis for layer II/III (Figure 5D) and mPFC (Figure 5E) neurons, and renewed Figure 5. The legend has been revised in accordance with this change.

2) There is no experimental proof that the analyzed dendrites were those that received dopaminergic innervation or affected in any way by the dopaminergic system.

We totally agree with the reviewer. We have replaced the sentence, “suggesting that both dopaminergic input into the mPFC and the reception by pyramidal neurons via dendritic spines are diminished in Del(3.0Mb)/+ mice”, with a milder expression (lines 710-711).

3) There are only 8 references (20%) from the past 5 years (6 references are from the 1990s), while most of the cited work is 10-20 years old. The list of references should be refreshed, if possible, and the discussion should refer to those newer findings. Here are some recent references that the authors should consider for inclusion and discuss in the manuscript; in general, these references contradict the finding that the microdeletion has no effect on the brain architecture:

Rogdaki M, Gudbrandsen M, McCutcheon RA, Blackmore CE, Brugger S, Ecker C, Craig MC, Daly E, Murphy DGM, Howes O. Magnitude and heterogeneity of brain structural abnormalities in 22q11.2 deletion syndrome: a meta-analysis. Mol Psychiatry. 2020 Aug;25(8):1704-1717. doi: 10.1038/s41380-019-0638-3. Epub 2020 Jan 10. PMID: 31925327; PMCID: PMC7387301.

Ching CRK, Gutman BA, Sun D, et al. Mapping Subcortical Brain Alterations in 22q11.2 Deletion Syndrome: Effects of Deletion Size and Convergence With Idiopathic Neuropsychiatric Illness. Am J Psychiatry. 2020;177(7):589-600. doi:10.1176/appi.ajp.2019.19060583

Kikinis Z, Cho KIK, Coman IL, et al. Abnormalities in brain white matter in adolescents with 22q11.2 deletion syndrome and psychotic symptoms. Brain Imaging Behav. 2017;11(5):1353-1364. doi:10.1007/s11682-016-9602-x

Padula MC, Schaer M, Armando M, et al. Cortical morphology development in patients with 22q11.2 deletion syndrome at ultra-high risk of psychosis. Psychol Med. 2018;48(14):2375-2383. doi:10.1017/S0033291717003920

Gudbrandsen M, Bletsch A, Mann C, et al. Neuroanatomical underpinnings of autism symptomatology in carriers and non-carriers of the 22q11.2 microdeletion. Mol Autism. 2020;11(1):46. Published 2020 Jun 8. doi:10.1186/s13229-020-00356-z

We have quoted 4 papers among the 5 that the reviewer recommended. These papers suggest substantial reductions in the cortical volumes and surface area in patients, which seem contradictory to our observations. We have added a discussion for this (lines 779-782).

4) It is curious that neither the mesocortical nor the mesolimbic pathways, presumed to be affected by the 22q11.2 deletion, was mentioned in the text.

Thank you for introducing the useful word. We have added the term, mesocortical pathway, to the end of section 3.6 (line 711). Generally, we would like to keep using “dopamine system”, because we did not confirm if the dopaminergic axons observed in the mPFC derive from the midbrain.

5) The microscopic figures should be made larger. They are very good, so why not show them in full bloom? For example, some details in Figure 1B, 2A, 3A, 4A are difficult to see. Similarly, some of the numbers on the graphs are almost impossible to see (Figure 4B, D, and those on the graphs in Figure 5).

We thank the reviewer for the helpful comment. We have revised these figures as the reviewer suggested. Figures 1-5 have been replaced in the draft manuscript.

Minor points:

6) Lines 76-77: The sentence should be changed to something that indicates that the number dendritic arborizations and/or spines was decreased…

Thank you for the helpful comment. We have corrected it accordingly.

7) Lines 87-90: Reference(s) should be given to explain why these primers were used (location of sequence, etc.)

We have added this information in lines 99-104.

8) Line 98: What is turboPFG? Please explain this abbreviation.

TurboRFP is a red fluorescent protein. We have added the information in line 113.

9) Line 106: mL? Impossible!

Thank you for pointing this out. That seems to be changed during the formatting. We have corrected it.

10) Line 114: The design and production of adeno-associated virus should be explained in more detail; please use appropriate references for each step.

We have added the detailed protocol of AAV production we conducted in the Methods.

11) Lines 128, 374: mL? Impossible!

Thank you for pointing them out. They seem to be changed during the formatting. We have corrected them.

12) Line 132: Embryonal age (E…) should be indicated. How was perfusion of the embryos performed? Please detail it.

We have provided the embryonic age we used and the detailed method to perfuse them.

13) Line 153: anti-chicken

We have corrected this (Chicken to chicken).

14) Lines 169-170: What type of “deep anesthesia” was used? It must be explained.

We used isoflurane. We have added this information.

15) Line 185: Fiji must be referenced or linked.

Actually, the link to Fiji was already provided in the previous section (section 2.7).

16) Line 256: The rationale to localize calbindin positive neurons is missing. Please provide references for the importance of calbindin positive interneurons.

We just used calbindin as a marker for interneurons. Anti-calbindin antibody is suitable for immunostaining in embryonic stages. We have revised this part in the manuscript and added references.

17) Line 308: the abbreviation EGFP at this fourth appearance is finally explained… Abbreviations should be spelled out when first used.

EGFP first appears in line 111. We have corrected this.

18) Line 371: Scale to Figure 4A is missing.

We have added the scale to Figure 4A. The legend has been changed accordingly.

19) Lines 375: The use of retro-orbital route should be referenced (for example, Yardeni et al., 2011). This paper states the correct amount of injectable fluid by this route.

Yardeni T, Eckhaus M, Morris HD, Huizing M, Hoogstraten-Miller S. Retro-orbital injections in mice. Lab Anim (NY). 2011 May;40(5):155-60. doi: 10.1038/laban0511-155. PMID: 21508954; PMCID: PMC3158461.

Thank you for recommending the reference regarding retro-orbital administration. We have quoted it in the Methods section.

20) Line 398: Correctly: “showing a”

We have corrected “showing the” to “showing an”.

21) Lines 452-458: As state above, the number of analyzed dendrites seems to be too small (for example, 8 dendrites from 3 brains). The number of dendrites analyzed gave a value that was significantly different between wild-type and Del(3.0Mb)/+ animals. Since this observation is one of the main findings of the study, a larger number of analyzed dendrites would have looked much better.

As we mentioned above, we have added the analysis for layer II/III (Figure 5D) and mPFC (Figure 5E) neurons, and renewed Figure 5. We hope this version is satisfactory.

22) Line 543: What is PERK? Why is it important in this study to reference it?

We have deleted the specific mention of PERK (line 790).

23) Lines 551-553: The authors hypothesize that “This might indicate the abnormal sensory input of Del(3.0 Mb)/+ mice, or more likely, the spine density may be reduced globally across the neocortex.” Are there any literature data to support any of these?

We observed reduced spine densities in layer II/III and layer V neurons in the somatosensory area. We discussed this in the first half of the sentence. As to the second half of the sentence, there are several lines of evidence showing lower intra-hemispheric connections in 22q11.2DS or its model mice including one paper the reviewer introduced in comment 3. We have added this explanation in lines 799-801.

24) Line 569: There was no Supplementary Materials available to this reviewer.

We are sorry for this inconvenience. We attached the Supplementary Figure below.

Round 2

Reviewer 1 Report

From my point of view, the manuscript could be accepted in the present form